# Fulminant Type 1 Diabetes Mellitus after SARS-CoV-2 Vaccination: A Case Report

**DOI:** 10.3390/vaccines10111905

**Published:** 2022-11-11

**Authors:** Rong Lin, Yu-Wei Lin, Mei-Hsiu Chen

**Affiliations:** 1Department of Internal Medicine, Division of General Internal Medicine, Far Eastern Memorial Hospital, New Taipei City 220, Taiwan; 2Department of Internal Medicine, Division of Endocrinology and Metabolism, Far Eastern Memorial Hospital, New Taipei City 220, Taiwan; 3Department of Medical Education, Division of General Practice, Far Eastern Memorial Hospital, New Taipei City 220, Taiwan; 4Department of Biomedical Engineering, Ming Chuang University, Taoyuan City 333, Taiwan

**Keywords:** diabetes mellitus, type 1, fulminant, SARS-CoV-2, SARS-CoV-2 vaccines

## Abstract

Severe acute respiratory syndrome coronavirus 2 (SARS-CoV-2) vaccines have been used worldwide to control the coronavirus disease pandemic. However, several adverse effects have been reported following vaccination. Therefore, further research on the adverse effects in individuals predisposed to life-threatening conditions is needed. Herein, we present a 39-year-old woman without any systemic disease who developed fulminant type 1 diabetes mellitus (T1DM) (low glycohemoglobin levels, despite hyperglycemia and diabetic ketoacidosis (DKA)) following SARS-CoV-2 vaccination. The patient was initially misdiagnosed as having fresh type 2 diabetes mellitus after the first episode of DKA, which was resolved by short-term insulin therapy and treated with oral anti-diabetic agents after the DKA was resolved. This made her develop a second episode of DKA shortly after treatment. The course and presentation of our case are noteworthy for alerting clinicians to vaccine-related fulminant T1DM.

## 1. Introduction

Type 1 diabetes mellitus (T1DM) typically occurs in young adulthood, with an age of onset between 15 and 29 years, and its incidence declines after puberty. The incidence of T1DM in the Asia-Pacific region is approximately 7/100,000 per year. T1DM is classified into type 1A and type 1B. Type 1A is autoimmune-related T1DM with several autoantibodies as biomarkers, whereas type 1B is idiopathic [1]. Fulminant T1DM is categorized as type 1B. As a rare subtype of T1DM, fulminant T1DM accounts for about 3% of cases of T1DM [2]. Risk factors of T1DM include viral infections, genetic factors, and environmental factors [1].

Several T1DM cases with a diverse age range have been reported following severe acute respiratory syndrome coronavirus 2 (SARS-CoV-2) vaccination [3,4,5,6,7,8,9]. This phenomenon is hypothesized to be caused by molecular mimicry-related cross-reactivity between SARS-CoV-2 antigens and receptors involved in autoimmunity [10] or autoimmune/inflammatory syndrome induced by adjuvants (ASIA) [11].

Herein, we present a rare case of SARS-CoV-2 vaccine-related fulminant T1DM, which rapidly progressed after symptom onset, and was characterized by a markedly low glycohemoglobin level.

We also reviewed other SARS-CoV-2 vaccine-related T1DM cases [3,4,5,6,7,8,9] for a better understanding of the clinical presentation of the vaccine related T1DM, in order to obtain a better treatment outcome in the future.

## 2. Case Description

A 39-year-old woman with a family history of type 2 diabetes mellitus (T2DM) visited our emergency department due to severe nausea and vomiting several times a day. The patient also reported dyspnea and palpitations. She had been discharged from another hospital 3 days previously after a 3-day hospitalization for newly diagnosed T2DM-related diabetic ketoacidosis (DKA). She denied having any systemic disease and medication history before this hospitalization. She had been treated with transient insulin infusion therapy, and the oral anti-diabetic agents, sitagliptin/metformin and gliclazide, had been prescribed. She denied having a SARS-Co-V-2 infection and had received two doses of the recombinant spike protein vaccine Medigen 8 months before the onset of symptoms, and a booster dose of the BNT162b2 mRNA-based vaccine (Pfizer-BioNTech) 14 weeks before admission.

On examination, the patient’s Glasgow Coma Scale score was E4V5M6. She had a slim build (body mass index: 19.2 kg/m^2^) and signs of acute illness. She had a body temperature of 36.8 °C, pulse rate of 95 beats/minute, respiratory rate of 18 breaths/minute, blood pressure of 108/59 mmHg, and oxygen saturation of 98% breathing ambient air. Laboratory tests revealed hyperglycemia and ketoacidosis (Table 1). Insulin pump therapy was initiated by the clinicians in the emergency department. After her blood glucose levels stabilized, the insulin pump was withdrawn, and a 4-split insulin regimen was initiated. Her hemoglobin A1c (HbA1c) level was 6.4%. These results led to a diagnosis of fulminant T1DM with ketoacidosis, and the 4-split insulin regimen (Tresiba 12u SC HS, Novorapid 10u SC TID) was maintained after discharge.

A glucagon test performed at the outpatient department on follow-up revealed a fasting C-peptide level of <0.020 ng/mL and a 6-min C-peptide level of 0.024 ng/mL. The levels of anti-glutamic acid decarboxylase and anti-islet antigen 2 (autoantibodies) were >2000 IU/mL and 1.150 U/mL, respectively. These results disclosed that the patient did not recover from T1DM and she needed insulin therapy to control her blood sugar. Finally, she received an adjusted dose of basal bolus insulin therapy (Tresiba 14u SC QD, Novorapid 5u,9u,9u SC) to control her blood sugar and kept her HbA1c to 6.4% three months after discharged.

## 3. Discussion

Fulminant T1DM was first described by Imagawa et al. [12]. The clinical manifestations of our patient met the criteria of fulminant T1DM, such as a rapid progression to DKA after the onset of hyperglycemic symptoms (≤7 days), relatively lower HbA1c levels (<8.5%) than serum glucose levels (>288 mg/dL), and stricter glucagon testing criteria (fasting C-peptide level of <0.3 ng/mL or post-glucagon stimulation C-peptide level of <0.5 ng/mL) [13].

If there is no insulin treatment for fulminant T1DM, the patient may die within one day; therefore, it is important to recognize the characteristic of the acute progression of fulminant T1DM early [14]. Thus, we have summarized literature on additional cases of new-onset T1DM after vaccination to provide a more comprehensive insight into this condition. According to current case reports of post-vaccination T1DM (Table 2), it tends to affect individuals aged 27−73 years, which is a later age of onset than that of non-iatrogenic T1DM. The reported symptoms include typical diabetic manifestations of polydipsia, polyuria, and polyphagia. Moreover, most of these cases (7/10) developed DKA. The symptom onset ranged from 3 days to 15 weeks after vaccination (median: 23.5 days). Moreover, most cases (9/10) in the literature were reported to occur following mRNA-based vaccination, with the exception of a case reported by Tang et al. [4] in a patient who had received an inactivated virus vaccine. The HbA1c levels varied from <6.5% to 16.3% (median: 9.7%). Seven cases indicated a positive autoimmunity correlation [3,6,9], and five of these cases occurred in individuals with T1DM-susceptibility, according to human leukocyte antigen (HLA) typing [3,4,5,6,7]. Only one case was defined as fulminant T1DM [5]; however, there were two more cases, although not stated, showing rapid onset T1DM with a lower HbA1c level [4,7]. Including our patients, four out of 11 vaccine-related T1DM were fulminant. It seems that vaccine related T1DM is more likely to be fulminant.

Risk factors of T1DM include viral infections (coxsackievirus B, enteroviruses, mumps, rubella, cytomegalovirus, and coronavirus); genetic factors, such as HLA DRB1-DQA1-DQB1 genotype; and environmental factors in countries with oceanic climates and less sunshine [1,15]. Vaccine-related insulin deficiency is hypothesized to be an outcome of the molecular mimicry of SARS-CoV-2 peptides in both the vaccine and human tissue antigens, which may cross-react and possibly result in autoimmunity. The presence of ASIA is another possible contributory factor, as adjuvants serve as immunological or pharmacological agents, thus enhancing the effectiveness of the vaccine by interfering with innate immunity and subsequently causing autoimmune disease in genetically susceptible individuals [16].

Talotta [16] discussed the correlation between mRNA-based vaccines and the incidence of vaccine-related T1DM and noted that pattern-recognition receptors and the dendritic cell-related cytokine pathway might be responsible for the unwanted autoimmune response. Pattern-recognition receptors, including toll-like receptors 3, 7, and 8, respond to RNA in endosomes and are possibly responsible for an ultimate inflammation response through type I interferon and transcription factor nuclear factor-kB. Taken together, mRNA-based vaccines may otherwise activate dendritic cells, subsequently affecting the downstream cytokine pathway. Although this enhances the immune response against SARS-CoV-2 infection, the overproduction of type I interferon may hinder the effectiveness of the vaccine as a result of the decreased translation of mRNA, thus causing the destruction of immune tolerance, and resulting in autoimmune reactions [16]. Apart from fulminant T1DM, vaccine-related endocrine systemic dysfunctions were also reported, such as Graves’ disease, subacute thyroiditis, hyperosmolar hyperglycemic stat, pituitary hypophysitis, adrenal hemorrhage, primary adrenal insufficiency, etc. [17].

Our patient had an older age of onset than that of typical T1DM, suggesting an idiopathic etiology. Owing to the possible autoimmune trigger and the later age of onset, our case is better classified as type 1B rather than type 1A, despite the presence of autoantibodies, which does not exclude the diagnosis of fulminant T1DM. Moreover, we observed a longer period from vaccination to T1DM onset. Although the patient had pre-existing risk factors for insidious and later-onset diabetes, it is possible that the T1DM occurred as a rare adverse effect of vaccination. Before the mRNA-based vaccine was taken, our patient had received two doses of a recombinant spike protein-based vaccine. We did not know if this combination contributed to the fulminant course of the vaccine related T1DM. Hence, more comprehensive research on T1DM development after SARS-CoV-2 vaccination of different mechanisms must be further conducted.

## 4. Conclusions

In conclusion, we reported that a woman developed middle-aged onset fulminant T1DM after SARS-CoV-2 vaccination. This is a rare and life-threatening adverse effect. Physicians should be aware of such clinical presentations, especially in patients who received SARS-CoV-2 vaccinations.

## Figures and Tables

**Table 1 vaccines-10-01905-t001:** Laboratory findings.

Parameter	Result	Reference Range
Venous blood gas (ambient air)		
pH	7.117	7.310–7.410
pCO_2_, mmHg	23.2	41.0–57.0
HCO_3_^−^, mmol/L	47.1	23.0–30.0
Base excess, mmol/L	−20.3	−2.0–2.0
Biochemistry		
Plasma glucose, mg/dL	364	70–100
Hb A1c, %	6.4	4.0–5.6
Plasma ketones, mmol/L	3.9	0.0–0.6
Fasting C-peptide, ng/mL	<0.020	1.100–4.400
6-min C-peptide, ng/mL	0.024	1.100–4.400
Free T4, ng/dL	0.648	0.80–2.00
Thyroid-stimulating hormone, µIU/mL	1.58	0.40–4.00
Hematology		
Hemoglobin, g/dL	14.3	12.0–16.0
Immunological tests		
Anti-GAD antibody, IU/mL	>2000	≤10.00
Anti-IA2 antibody, U/mL	1.15	≤7.00
Infection		
SARS-CoV-2 PCR	Negative	Negative

Anti-GAD, anti-glutamic acid decarboxylase; anti-IA2, anti-islet antigen 2; Hb A1c, glycosylated hemoglobin; HCO_3_^−^, bicarbonate ion; pCO_2_, partial pressure of carbon dioxide; PCR, polymerase chain reaction; SARS-CoV-2, severe acute respiratory syndrome coronavirus 2; T4, thyroxine.

**Table 2 vaccines-10-01905-t002:** Relevant case reports of post-vaccination type 1 diabetes mellitus.

	Author	Age (Years), Sex	Vaccine	Onset	HbA1c (%)	BMI (kg/m^2^)	Auto ab.	Glucagon Test	T1DM HLA Typing
Case 1	Yano et al. [3]	50, M	M × 2	28 days	10.3	18.3	Insulin autoab.	(+)	(+)
Case 2	Tang et al. [4]	45, F	CoronaVac	6 days	Wnl.	18.1	(−)	(+)	(+)
Case 3 ^†^	Sasaki K et al. [5]	73, F	B × 2	6 days	7.6	20.6	(−)	(+)	(+)
Case 4	Sasaki H et al. [6]	36, F	M × 2	8 weeks	9.3	N/A	Insulin autoab., anti-GAD	(+)	(+)
Case 5	Sakurai et al. [7]	36, F	B × 2	3 days	7.0	N/A	(−)	(+)	(+)
Case 6	Bleve et al. [8]	61, F	B × 2	26 days	11.5	N/A	N/A	N/A	N/A
Case 7	Aydoğan et al. [9]	56, M	B × 2	15 weeks	8.2	27.4	Anti-GAD	(−)	N/A
Case 8	Aydoğan et al. [9]	48, M	B × 2	8 weeks	10.1	21.9	Anti-GAD	(−)	N/A
Case 9	Aydoğan et al. [9]	27, F	B × 2	3 weeks	12.5	20.0	Anti-GAD	(−)	N/A
Case 10	Aydoğan et al. [9]	36,M	CoronaVac × 2, B × 2	3 weeks	12.6	22.8	Anti-GAD	(+)	N/A
Our case ^†^	Lin et al.	39, F	Medigen × 2, B	14 weeks	6.4	19.2	Anti-GAD, anti-IA-2	(+)	N/A

^†^ Fulminant T1DM. Vaccine abbreviations: B, BNT162b2; M, mRNA-1273. Other abbreviations: BMI, body mass index; GAD, glutamic acid decarboxylase; IA-2, islet antigen 2; N/A, not available; T1DM HLA, type 1 diabetes mellitus human leukocyte antigen susceptibility typing.

## Data Availability

All of the relevant data are provided in the paper.

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
