# Peer review of "Fulminant Type 1 Diabetes Mellitus after SARS-CoV-2 Vaccination: A Case Report"

_vaccines, 2022, doi:10.3390/vaccines10111905_

Round 1

Reviewer 1 Report

Please, carry on reporting and documenting adverse reactions following nucleic acid-based vaccination, which neither prevents infection nor disease spread.

Author Response

Thank you for your comment.

Reviewer 2 Report

This is an interesting topic discussing a fulminant complication following covid19 vaccine. 

1. The time interval between vaccination and the development of DKA looked to be too long. how do you prove the association between T1DM and covid vaccination? there is only 1 case report (case 7 in table 2) suggesting late development of type I DM following the covid vaccine.

2.  what happens after this event? Did she recover from type 1 DM or she needed to be treated with insulin for a long-time? You need to describe the following clinical course of this patient.

Usually, the effect of covid vaccine does not last longer than 6 months.   I am wondering about the clinical course of this patient and other case reports.

3. Did you follow the autoantibodies of this patient?

4. what are the mechanisms involved in the development of fulminant type 1DM following vaccination?

5. Do you think that the mixed vaccination affected the development of fulminant type 1 DM in this patient? what will be the underlying mechanism?

-.> need to be described in the discussion

6. If this complication is associated with autoimmunity accelerated by the covid vaccine, did you try immune-suppressive treatment for this patient?

and what about other case reports? 

Author Response

Author's Reply to the Review Report (Reviewer 2)

  1. The time interval between vaccination and the development of DKA looked to be too long. how do you prove the association between T1DM and covid vaccination? there is only 1 case report (case 7 in table 2) suggesting late development of type I DM following the covid vaccine.

Ans: We agreed with your comment, we cannot prove the association between T1DM and covid vaccination, more comprehensive research on T1DM after SARS-CoV-2 vaccination of different mechanisms must be further conducted.

  1. what happens after this event? Did she recover from type 1 DM or she needed to be treated with insulin for a long-time? You need to describe the following clinical course of this patient.

Usually, the effect of covid vaccine does not last longer than 6 months.   I am wondering about the clinical course of this patient and other case reports.

Ans: After this event, the patient does not recover from type 1 DM, she still needs basal-bolus insulin therapy (Tresiba 14u QD and Novorapid 5u, 9u, 9u TID ) to control her blood sugar and keep her HbA1c to 6.4% three months after discharged in out-patient follow-up.

  1. Did you follow the autoantibodies of this patient?

Ans: We checked the autoantibodies one month after the patient recovered from diabetic ketoacidosis and the title of Anti-GAD Ab was >2000.00 IU/mL and type 1 diabetes was diagnosed, so we think it is not necessary to follow the autoantibodies.  

  1. what are the mechanisms involved in the development of fulminant type 1DM following vaccination?

Ans: We added the paragraphs below in discussion.

Risk factors of T1DM include viral infections (coxsackievirus B, enteroviruses, mumps, rubella, cytomegalovirus, and coronavirus); genetic factors such as HLA DRB1-DQA1-DQB1 genotype; and environmental factors in countries with oceanic climates and less sunshine [1,15]. Vaccine-related insulin deficiency is hypothesized to be an outcome of the molecular mimicry of SARS-CoV-2 peptides in both the vaccine and human tissue antigens, which may cross-react and possibly result in autoimmunity. The presence of ASIA is another possible contributory factor, as adjuvants serve as immunological or pharmacological agents, thus enhancing the effectiveness of vaccine by interfering with innate immunity and subsequently causing autoimmune disease in genetically susceptible individuals [16].

Talotta [16] discussed the correlation between mRNA-based vaccines and the incidence of vaccine-related T1DM and noted that pattern-recognition receptors and the dendritic cell-related cytokine pathway might be responsible for the unwanted autoimmune response. Pattern-recognition receptors, including toll-like receptors 3, 7, and 8, respond to RNA in endosomes and are possibly responsible for an ultimate inflammation response through type I interferon and transcription factor nuclear factor-kB. Taken together, mRNA-based vaccines may otherwise activate dendritic cells, subsequently affecting the downstream cytokine pathway. Although this enhances the immune response against SARS-CoV-2 infection, overproduction of type I interferon may hinder the effectiveness of the vaccine as a result of decreased translation of mRNA, thus causing destruction of immune tolerance, and resulting in autoimmune reactions [16]. Except fulminant T1DM, vaccine-related endocrine systemic dysfunctions were also reports, such as Graves’ disease, subacute thyroiditis, hyperosmolar hyperglycemic stat, pituitary hypophysitis, adrenal hemorrhage, primary adrenal insufficiency, etc.

  1. Do you think that the mixed vaccination affected the development of fulminant type 1 DM in this patient? what will be the underlying mechanism?

-.> need to be described in the discussion

Ans: As mentioned above, vaccination may trigger autoimmunity, not limited to mixed vaccination.

  1. If this complication is associated with autoimmunity accelerated by the covid vaccine, did you try immune-suppressive treatment for this patient?

Ans: No, we did not try immune-suppressive treatment for this patient.

and what about other case reports?

They did not try immune-suppressive treatment as well.

Reviewer 3 Report

Summary of the Work

This work reports a case of a woman of middle-aged onset fulminant type 1 diabetes mellitus (T1DM) developing after SARS-CoV-2 vaccination, which rapidly progressed after symptom onset, and was characterized by a markedly low glycohemoglobin level. Other cases of SARS-CoV-2 vaccine-related T1DM were also briefly reviewed.

General Considerations

- The case of (fulminant) T1DM developed after SARS-CoV-2 vaccination is clearly described with the help of accurate laboratory findings.

- The work is interesting and topical. However, as pointed out by the authors (see Table 2.), numerous works on this subject have already appeared recently in the literature.

- The list of the cited references should be completed taking into account the results that recently appeared in the literature on this topic.

Suggestions

1) The fact that COVID-19 is linked to diabetes is already well-established and widely reported in the literature. Statistics show that people who recovered from COVID-19 were 28% more likely to develop diabetes in the months afterward. In particular, some people with both type 1 diabetes (T1D) and type 2 diabetes (T2D) have experienced brief blood sugar spikes after receiving one of the doses. According to the current interpretation, people having pre-diabetes with chronic, subclinical inflammation before getting COVID-19, when they do get infected with the virus, may develop hyperinflammation that causes metabolic stress and more insulin resistance. Consequently, they manifest diabetes. The authors should highlight more precisely the added value of this work, highlighting its innovative aspects, compared to those that have recently appeared on this research topic.

2) Johns Hopkins doctors conducted a research study to see if the COVID-19 booster vaccine increases glucose levels or insulin needs in patients with type 1 diabetes. Specifically, the study was conducted on patients who received two Pfizer/Modern COVID-19 vaccinations or one J&J vaccination. The results described hyperglycemia and/or insulin resistance following the administration of the COVID-19 vaccine (although further investigations into this matter are needed). Authors are asked to comment on this (i.e., the link between SARS-CoV-2 vaccination and insulin resistance).

3) Dysglycemia in type 1 or 2 DM is increasingly reported following COVID-19 infection or vaccination, but the rapid progression of prediabetes to type 1 diabetes mellitus has rarely been reported. Do the authors have more precise data on this?

4) The authors simply concluded that "this is a rare and life-threatening adverse effect". Written in this form, this may be a source of objections. More specifically, to vaccination against SARS-CoV2 infection, what do the authors suggest as an alternative strategy to apply to patients with pre-diabetes to Type 1 diabetes mellitus.? Restrictive measures?

Conclusions

We have to mention that it is already commonly accepted by the scientific community that COVID can have a (dramatic) effect on the stimulation of cytokines, which are inflammatory factors. This worsens insulin resistance and can raise blood sugar. In the literature, we can find several studies showing that dysglycemia in Type 1 or 2 DM is increasingly reported after COVID-19 infection or vaccination. So, as said, the topic treated in this study is undoubtedly interesting and topical, but the authors are invited to highlight the added value of their findings and suggest, if possible, alternative treatments to prediabetes to type 1 diabetes mellitus.

Author Response

Author's Reply to the Review Report (Reviewer 3)

Summary of the Work

This work reports a case of a woman of middle-aged onset fulminant type 1 diabetes mellitus (T1DM) developing after SARS-CoV-2 vaccination, which rapidly progressed after symptom onset, and was characterized by a markedly low glycohemoglobin level. Other cases of SARS-CoV-2 vaccine-related T1DM were also briefly reviewed.

General Considerations

- The case of (fulminant) T1DM developed after SARS-CoV-2 vaccination is clearly described with the help of accurate laboratory findings.

- The work is interesting and topical. However, as pointed out by the authors (see Table 2.), numerous works on this subject have already appeared recently in the literature.

- The list of the cited references should be completed taking into account the results that recently appeared in the literature on this topic.

Suggestions

1) The fact that COVID-19 is linked to diabetes is already well-established and widely reported in the literature. Statistics show that people who recovered from COVID-19 were 28% more likely to develop diabetes in the months afterward. In particular, some people with both type 1 diabetes (T1D) and type 2 diabetes (T2D) have experienced brief blood sugar spikes after receiving one of the doses. According to the current interpretation, people having pre-diabetes with chronic, subclinical inflammation before getting COVID-19, when they do get infected with the virus, may develop hyperinflammation that causes metabolic stress and more insulin resistance. Consequently, they manifest diabetes. The authors should highlight more precisely the added value of this work, highlighting its innovative aspects, compared to those that have recently appeared on this research topic.

Although COVID-19 infection worsens the glycemic control in patients with diabetes is well-established, it is not applied to our patient.

2) Johns Hopkins doctors conducted a research study to see if the COVID-19 booster vaccine increases glucose levels or insulin needs in patients with type 1 diabetes. Specifically, the study was conducted on patients who received two Pfizer/Modern COVID-19 vaccinations or one J&J vaccination. The results described hyperglycemia and/or insulin resistance following the administration of the COVID-19 vaccine (although further investigations into this matter are needed). Authors are asked to comment on this (i.e., the link between SARS-CoV-2 vaccination and insulin resistance).

Although COVID-19 vaccination may increase insulin resistance in patients with type 2 DM or increase insulin needs in patients with type 1 DM, it is not applied to our patient.

3) Dysglycemia in type 1 or 2 DM is increasingly reported following COVID-19 infection or vaccination, but the rapid progression of prediabetes to type 1 diabetes mellitus has rarely been reported. Do the authors have more precise data on this?

Developing Type 1 DM after COVID-19 vaccination is rare, and there are only 11 cases (including ours) reported. The mechanism involved is not clear. There are some proposed inferences and we have added in our article with the followings:

Risk factors of T1DM include viral infections (coxsackievirus B, enteroviruses, mumps,

rubella, cytomegalovirus, and coronavirus); genetic factors such as HLA DRB1-DQA1-

DQB1 genotype; and environmental factors in countries with oceanic climates and less

sunshine [1,15]. Vaccine-related insulin deficiency is hypothesized to be an outcome of

the molecular mimicry of SARS-CoV-2 peptides in both the vaccine and human tissue

antigens, which may cross-react and possibly result in autoimmunity. The presence of

ASIA is another possible contributory factor, as adjuvants serve as immunological or

pharmacological agents, thus enhancing the effectiveness of vaccine by interfering with

innate immunity and subsequently causing autoimmune disease in genetically susceptible

individuals [16].

Talotta [16] discussed the correlation between mRNA-based vaccines and the incidence

of vaccine-related T1DM and noted that pattern-recognition receptors and the dendritic

cell-related cytokine pathway might be responsible for the unwanted autoimmune

response. Pattern-recognition receptors, including toll-like receptors 3, 7, and 8, respond

to RNA in endosomes and are possibly responsible for an ultimate inflammation response

through type I interferon and transcription factor nuclear factor-kB. Taken together,

mRNA-based vaccines may otherwise activate dendritic cells, subsequently affecting the

downstream cytokine pathway. Although this enhances the immune response against

SARS-CoV-2 infection, overproduction of type I interferon may hinder the effectiveness

of the vaccine as a result of decreased translation of mRNA, thus causing destruction of

immune tolerance, and resulting in autoimmune reactions [16]. Except fulminant T1DM,

vaccine-related endocrine systemic dysfunctions were also reports, such as Graves’

disease, subacute thyroiditis, hyperosmolar hyperglycemic stat, pituitary hypophysitis,

adrenal hemorrhage, primary adrenal insufficiency, etc.

4) The authors simply concluded that "this is a rare and life-threatening adverse effect". Written in this form, this may be a source of objections. More specifically, to vaccination against SARS-CoV2 infection, what do the authors suggest as an alternative strategy to apply to patients with pre-diabetes to Type 1 diabetes mellitus.? Restrictive measures?

It is almost impossible to prevent type 1 DM because it is an organ-specific autoimmune disease caused by the autoimmune response against pancreatic β cells. However, timely insulin administration is enough to control the acute complications, such as diabetic ketoacidosis. Thus, report the possible adverse effects after vaccination would alert us about the hazard and help us to control the adverse effects as soon as possible.  

Conclusions

We have to mention that it is already commonly accepted by the scientific community that COVID can have a (dramatic) effect on the stimulation of cytokines, which are inflammatory factors. This worsens insulin resistance and can raise blood sugar. In the literature, we can find several studies showing that dysglycemia in Type 1 or 2 DM is increasingly reported after COVID-19 infection or vaccination. So, as said, the topic treated in this study is undoubtedly interesting and topical, but the authors are invited to highlight the added value of their findings and suggest, if possible, alternative treatments to prediabetes to type 1 diabetes mellitus.

As we mentioned above, it is almost impossible to prevent type 1 DM because it is an autoimmune disease. Although we are not able to prevent it, we can control it by giving insulin timely. Thus, report the possible adverse effects after vaccination would alert us about the hazard and help us to control the adverse effects as soon as possible.

Reviewer 4 Report

Dear Author,

This is an interesting paper. I agree with your approach of this case in relationship to the vaccination, not necessarily based on the level of statistical evidence we have so far in literature, but based on the importance of awareness in such severe cases. Also, the scientific evidence amid COVID-19 is expected to increase thus any scientific documented report matters.

Here are my observations/questions/comments:

1.    Abstract   - It is not clear “initially diagnosed as type 2..” meaning after vaccine or during her personal medical records regardless the timing of vaccination (Of course, you explained it in the main text body, but it should be clear in Abstract, too.)

2.    Introduction – I suggest a brief statement concerning other specific risk factors with respect to fulminant type 1 diabetes mellitus since post-vaccine circumstances are new among this panel

3.    Case report – medical history – Did the patient had COVID-19 infection among her medical records at any point?

4.    Case report – personal medical background – Did the patient had any personal history of autoimmune conditions, including reactions to drugs or other types of vaccines?

5.    Case report – outcome and further management – Do you have any data with regard to the follow-up of the patient? Which is the overall prognostic and long term management/recommendations including check-ups?

6.    Discussion – Which is the mortality rate in these cases (with/without insulin therapy?

7.    Discussion - Do you consider the vaccination as a trigger of a medical condition that might have been present anyway at some point?

8.    Discussion - Do you consider that, in case the patient was not initially treated as type 2 diabetes mellitus, the post-vaccination outcome would have been different?

9.    Discussion – I suggest you also introduce a brief comment on other fulminant reactions reported after COVID-19 vaccine (of endocrine and metabolic type, etc.)

Best regards,

Round 2

Reviewer 2 Report

Most of the comments were well-addressed.

Reviewer 3 Report

The authors answered, albeit in a partial way, the questions raised in my first report (in particular, answers to questions 2 and 4, considered together, leave me somewhat unsatisfied).  Furthermore, the results reported in this study refer to a single patient. It is not clear whether the findings found in this study are of general validity or refer to a quite rare event. Anyhow, I agree with the authors that reporting possible adverse effects after vaccination would alert us and help us to control the adverse effects as soon as possible. For this reason, in my opinion, the manuscript deserves to be published.